# Smaller infarct size with ticagrelor vs. clopidogrel in STEMI patients: Insights from cardiac magnetic resonance

Francisco A Fonseca [1]*, Adriano Caixeta[1], Gilberto Szarf[1,2], Ibraim Pinto[3], Antonio M Figueiredo Neto [4], Carolina N França[5], Henrique T Bianco [1], Henrique A Fonseca[2], Amanda S Bacchin[5], Michelle Birtche[1], Igor R M Batista [1], Maria C Izar [1,6], for the BATTLE-AMI investigators[¶]

1 Cardiology Division, Universidade Federal de São Paulo, São Paulo, Brazil, 2 Hospital Israelita Albert Einstein, São Paulo, Brazil, 3 Instituto Dante Pazzanese de Cardiologia, São Paulo, Brazil, 4 Instituto de Física, Universidade São Paulo, São Paulo, Brazil, 5 Universidade Santo Amaro, São Paulo, Brazil, 6 Brazilian Network of Collaboration and Knowledge Advancement on Severe Hypertriglyceridemia – Hypertri Brazil Network, Casa Dos Raros, Porto Alegre, Brazil,

¶ The B And T Types of Lymphocytes Evaluation in Acute Myocardial Infarction (BATTLE-AMI) trial investigators are listed in the appendix of the supplementary file
* fah.fonseca@unifesp.br

## Abstract

### Background

Ticagrelor has many protective cardiovascular properties beyond potent antiplatelet action. This study aimed to compare the effects of ticagrelor versus clopidogrel on infarcted mass, quantified by cardiac magnetic resonance (CMR), in patients with ST-segment elevation acute myocardial infarction (STEMI).

### Methods

Adult patients of both sexes with STEMI under a pharmaco-invasive strategy were included (n = 225). Patients were treated by thrombolysis within six hours of symptom onset and underwent angiography with percutaneous coronary interventions, when needed, within the first 24 hours. Prior to the invasive procedures, patients were randomly assigned to receive either ticagrelor or clopidogrel using a centralized computerized system. Patients were followed on a weekly basis to optimize their medical therapy.

### Results

After 30 days, CMR was performed and a smaller percentage of left ventricular infarcted mass was found with ticagrelor (p = 0.012), despite similar angiographic findings at baseline (Syntax score, Gensini score, culprit artery, TIMI flow, and myocardial blush). At 30 days, left ventricular ejection fraction (LVEF) was

**Data availability statement:** The dataset was activated at the DRYAD site, it is available at (https://doi.org/10.5061/dryad.8pk0p2p1v).

**Funding:** This study received financial support from the São Paulo Research Foundation (FAPESP, grant 2012/51692-7) and through an investigator-initiated grant from AstraZeneca (ESR 14-10726). The following agencies provided additional funding: INCT /CNPq (Conselho Nacional de Desenvolvimento Científico e Tecnológico; grants 465259/2014-6 and 303001/2019-4), INCT/FAPESP (grant 14/50983-3), INCT/CAPES (Coordenação de Aperfeiçoamento de Pessoal de Nível Superior; grant 88887.136373/2017-00), FAPESP (Thematic Project; grant 2016/24531-3), and INCT-FCx (Instituto Nacional de Ciência e Tecnologia de Fluidos Complexos). None of the sponsors influenced the study design, data collection, interpretation, and or publications.

**Competing interests:** No authors have competing interests.

comparable between groups. Still, the K-means algorithm displayed more homogeneous responses for smaller infarcted mass and better LVEF among those patients treated with ticagrelor. Standard lipid panel and most inflammatory parameters were similar at baseline and after 30 days. However, lower high-sensitivity troponin T and high-sensitivity C-reactive protein levels were found in samples collected from patients treated with ticagrelor on the first day of STEMI.

## Conclusion

In patients with STEMI under a pharmaco-invasive strategy, therapy with ticagrelor was associated with a smaller infarct size than clopidogrel.

## Trial registration

Clinicaltrials.gov (NCT02428374).

---

## Introduction

Even more than a decade after the PLATO study [1], the reasons behind the superior clinical outcomes of ticagrelor compared to clopidogrel remain unclear. Improvements in microcirculation and enhanced antiplatelet activity have been suggested [2,3], but the advantages of ticagrelor over clopidogrel remain controversial [4–6]. Recently, interest in ticagrelor has been renewed due to excellent outcomes as monotherapy in patients undergoing percutaneous coronary intervention (PCI) after an initial period of dual antiplatelet therapy [7,8]. Additionally, in patients aged 75 years or younger, bleeding rates have been comparable to those observed with clopidogrel, even among patients receiving thrombolytic therapy [9,10].

Currently, residual lipid [11,12], inflammatory [13–15], and thrombotic [16–18] risks have been suggested as key factors contributing to the high rate of recurrent events following acute myocardial infarction [19]. Furthermore, cardiac magnetic resonance (CMR) provides an accurate assessment of infarct size and ventricular remodeling after acute myocardial infarction, offering valuable prognostic insights [20].

In many countries, most myocardial infarctions are initially managed in hospitals lacking 24-hour access to hemodynamic services. Consequently, when timely PCI is not feasible, the pharmaco-invasive strategy emerges as the most practical and effective therapeutic approach [21–23].

This prospective, randomized clinical trial aimed to compare the effects of ticagrelor versus clopidogrel on infarct size, as assessed by CMR, in patients with ST-segment elevation myocardial infarction (STEMI) undergoing a pharmaco-invasive treatment strategy.

## Methods

The CONSORT checklist is available as S3 File.

## Ethics approval

The study protocol was approved by the local ethics committee (IRB 0297/2014; CAAE: 71652417.3.0000.5505), which follows the last version of the Helsinki Declaration. All patients signed the informed consent prior to any study procedure.

## Study design

The comparison between ticagrelor and clopidogrel in patients with STEMI is part of a thematic study with pre-specified analyses that also included two lipid-lowering strategies. The study design, including objectives and inclusion/exclusion criteria, was previously published [24]. Briefly, the BATTLE-AMI study (NCT02428374) is a prospective, randomized, open-label clinical trial with blinded CMR and angiographic analyses. Adult STEMI patients who underwent thrombolytic therapy were included. After providing written informed consent, eligible patients were randomized in a 1:1 ratio to receive either ticagrelor or clopidogrel using a centralized computerized system before coronary angiography.

## Patient Recruitment and Treatment

Consecutive STEMI patients of both sexes, aged 18–75 years, were treated by tenecteplase (Metalyse®, Boehringer Ingelheim) in the first 6 hours of symptom onset, according to patients' weight, and received 300 mg of clopidogrel and 300 mg of aspirin in public hospitals of the city of São Paulo, Brazil, as part of the SP STEMI treatment network [24]. In the network routine, patients were subsequently transferred to a tertiary teaching hospital and underwent systematic early invasive coronary angiography. Thus, the patients of the study were referred to our university hospital in the first 24 hours, and randomly assigned 1:1 to either ticagrelor (Brilinta®, Astrazeneca) 90 mg b.i.d. after a loading dose of 180 mg or clopidogrel (Plavix®, Sanofi) 75 mg q.d. with an additional loading dose of 300 mg of clopidogrel at the discretion of the attending physician [24]. Coronary angiography was performed in all patients in the first 24 hours and PCI, when needed, in the culprit lesion. Additional coronary interventions in non-culprit lesions were performed at the same time or electively according to the hemodynamic team's decision. All these interventions conform to the recommended guidelines [25]. As part of the BATTLE-AMI study [24], patients were randomized to receive rosuvastatin 20 mg (Crestor®, Astrazeneca) or simvastatin 40 mg combined with ezetimibe 10 mg (Vytorin®, MSD). Patients with clinical instability, previous myocardial infarction or coronary revascularization, severe renal insufficiency, active liver disease, immune diseases, or known intolerance to the study drugs were excluded. Thirty days after STEMI, a CMR was performed. Patients were included from May 2015 to March 2020. All laboratory, angiographic, or CMR analyses were performed blindly.

## Laboratory analysis

Blood samples were collected on the first day, or in the morning of the next day, in case of overnight hospital admission, and also after 30 days.

Routine laboratory parameters were performed in the central laboratory of the university hospital. B and T Lymphocyte subtypes were examined by flow-cytometry as previously reported [26]. Circulating cytokines were determined by enzyme-linked immunosorbent assay (ELISA) [27].

## Quantitative coronary angiography

An independent angiographic core laboratory assessed quantitative coronary angiography blinded to randomization assignment and clinical outcomes for baseline and post-PCI lesions with the use of validated quantitative methods (CMS; Medis Medical Imaging Systems, Leiden, The Netherlands) [28,29]. The path line, vessel contour, lumen, reference diameters, and lesion length were determined automatically by the contrast density, occasionally requiring editing by analysts. The following parameters were obtained through quantitative coronary angiography: (i) minimal lumen diameter; (ii) reference vessel diameter; (iii) obstruction length; (iv) and percent diameter stenosis. Rentrop score, TIMI flow, myocardial

blush, and the corrected TIMI frame count grades in the infarcted vessel were defined as previously reported [30–32]. Each lesion with ≥50% diameter stenosis in vessels ≥1.5 mm was scored using the syntax score algorithm (https://syntax-score.org/), fully described elsewhere [33].

### Cardiac magnetic resonance

In 3.0 T clinical scanners (Siemens, Erlangen, Germany, Philips) patients remained in the supine position, and a phased-array receiver cardiac coil was placed on the chest. Cine images were ECG-gated and were acquired with approximately 8 sec breath holds for each slice. Cine images were acquired in the short-axis view (from the mitral-valve insertion plane through the left ventricle until the apex) and three long-axis views. Slice thickness was 8 mm, and slice spacing was 2 mm. All indexes studies were done with the injection of a commercially available gadolinium-based contrast agent (gadopentetate dimeglumine or gadoteridol) administered intravenously at a dose of 0.2 mmol per kilogram of body weight, and a dynamic, breath-hold MR first-pass perfusion examination were obtained and resulted in three short-axis slices of 8–10 mm thickness were acquired every heart beat along the long axis, with spatial resolution of 2–3 × 2–3 mm. Stress data were acquired following 6 min of dipyridamole (0.56 mg/min/kg intravenous), followed by an aminophylline injection to revert hyperemia. A rest injection of the same contrast agent (0.2 mmol per kilogram of body weight) was done to obtain images that would be compared with those of the stress phase. After a 5-minute delay to allow for washout of the contrast agent from healthy myocardium, magnitude and phase-sensitive inversion recovery images following an inversion recovery pulse were recorded to allow the detection of late gadolinium enhancement. These were registered in the same views used for cine MRI and used to assess for myocardial viability. The inversion time suggested by the protocol could be adjusted by the technologist to optimize image quality. Typical voxel size was 1.9 by 1.4 by 8 mm. Scanning protocol for the follow-up study included segmented cine CMR (short and long axis views) for cardiac function, followed by segmented late gadolinium enhancement imaging for viability. Myocardial infarction size was quantified by manual contouring of the late gadolinium enhancement areas in each of the short axis slices, using dedicated software, as described elsewhere [34].

### Study endpoints and predictor definitions

The primary objective of the study was to compare the amount of infarcted myocardial mass measured by CMR after 30 days of treatment with ticagrelor or clopidogrel, among STEMI patients under the pharmaco-invasive strategy. A secondary objective was the left ventricular ejection fraction (LVEF). Circulating inflammatory cells and biochemical variables were also examined.

### Statistical analysis

All analyses were made using software R Core Team (2024), and the level of significance was stated at 5%. Data was reported as median and interquartile range or mean and standard deviation. Qualitative variables were reported as numbers and percentages. The Shapiro-Wilk test was used to examine normality. The Mann-Whitney test was used to compare nonparametric variables for independent groups, including myocardial fibrosis (% and g), and LVEF. Categorical variables were compared by the Pearson's Chi-square test or Fisher's exact test. Correlations between variables were examined by the Spearman's Rho test. The K-means algorithm was used for cluster analysis involving LVEF and myocardial fibrosis. Sensitivity analysis was conducted for infarcted mass and LVEF using Multivariate Imputation by Chained Equations in the R software [35].

## Results

Fig 1 shows the study flow chart. A total of 286 patients were assessed for eligibility and from those enrolled in the trial only few patients did not complete the trial or did not perform the CMR studies.

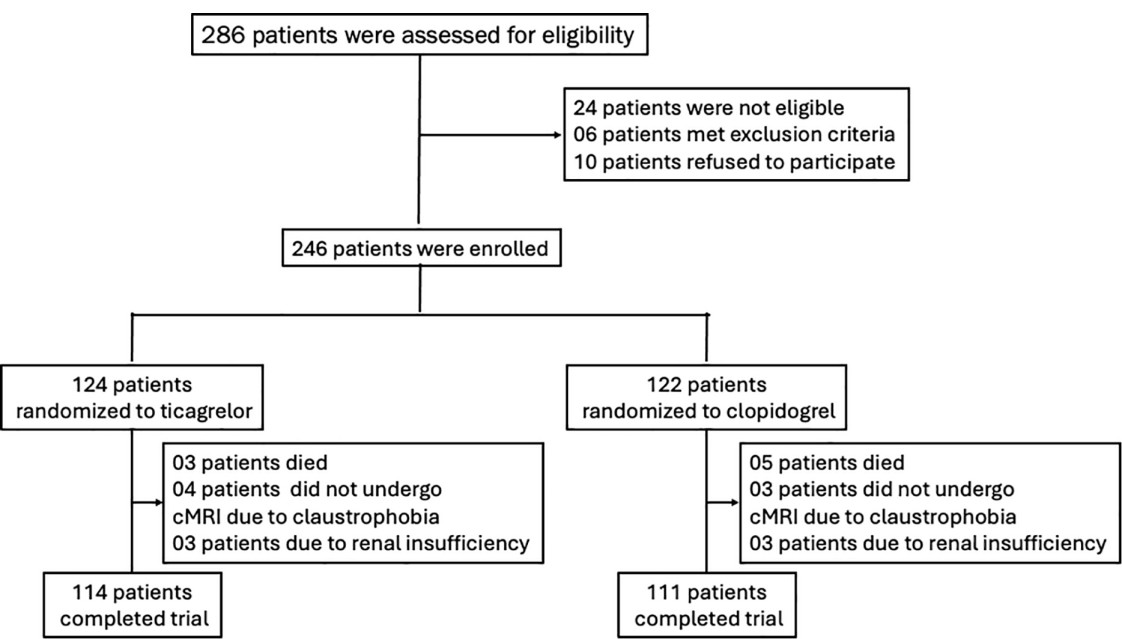

**Fig 1. Study flowchart.** Of the 286 patients assessed, 40 were not enrolled in the trial due to inclusion/exclusion criteria or refusal to participate. Among those randomized to ticagrelor or clopidogrel, 21 did not complete the trial.

## Baseline Characteristics

The study population was composed mainly of middle-aged men, a quarter of whom had diabetes and were smokers (Table 1). Among laboratory parameters, higher levels of high-sensitivity troponin T (hsTNT) and high-sensitivity C-reactive protein (hsCRP) were observed among patients randomized to the clopidogrel arm, on the first day after hospitalization (Fig 2). All patients were followed weekly to receive optimal medical therapy (91% received beta-blockers, 97% ACEI or ARB, 100% statins monotherapy or combined with ezetimibe), and no differences were noted in medical therapy between antiplatelet arms. The median (IQR) time from onset of symptoms to pharmacological thrombolysis was similar between groups [ticagrelor 180 (120–314) minutes; clopidogrel 180 (120–240) minutes; p = 0.31, Mann Whitney U test] and the median time from thrombolysis to percutaneous coronary intervention was 17 (7–24) hours for ticagrelor and 12 (7–25) hours for clopidogrel, without differences between the arms of the study (p = 0.98, Mann-Whitney test). The same number of patients were referred for rescue angioplasty in both arms, but only stable patients were included in the trial.

## Laboratory parameters

After 30 days, no differences between antiplatelet arms were observed for the standard lipid panel (total cholesterol, LDL-C, HDL-C, triglycerides, non-HDL-C), as well as for glucose, creatinine, or inflammatory variables, including lymphocyte subtypes and interleukins (Table 2 and S1 File). There were no differences between groups regarding the lipid-lowering therapies assigned for each antiplatelet group (Chi-square test, p = 0.69).

## Angiographic findings and PCI details

Coronary angiography and percutaneous coronary intervention were performed during the first 24 hours of STEMI. All patients received the randomized antiplatelet therapy before the hemodynamic procedures. The predominant culprit coronary arteries were the left anterior descending artery (LAD) or right coronary artery (RCA) and, less frequently left

**Table 1. Characteristics of study population at baseline.**

| | Ticagrelor N = 114 | Clopidogrel N = 111 | P value |
|---|---|---|---|
| Age, years | 57 (52-63) | 56 (50-62) | 0.59 |
| Male gender | 75 (66) | 85 (77) | 0.07 |
| Hypertension | 35 (31) | 46 (41) | 0.10 |
| Diabetes | 29 (25) | 28 (25) | 0.47 |
| Smoking | 26 (23) | 27 (24) | 0.60 |
| BMI, kg/m$^2$ | 26.9 (24.4-29.6) | 26.6 (24.2-29.7) | 0.99 |
| SBP, mm Hg | 128 (111-137) | 125 (110-140) | 0.58 |
| DBP, mm Hg | 75 (70-89) | 80 (70-87) | 0.97 |
| hsTNT, ng/L | 4027 (1652-9406) | 6177 (2233-11479) | 0.03 |
| HbA1c, % | 5.9 (5.6-6.5) | 5.9 (5.5-6.5) | 0.25 |
| Glucose, mg/dL | 118 (101-139) | 121 (100-163) | 0.30 |
| Total cholesterol, mg/dL | 199 (173-237) | 195 (169-225) | 0.48 |
| LDL-cholesterol, mg/dL | 128 (105-154) | 129 (106-159) | 0.69 |
| HDL-cholesterol, mg/dL | 42 (34-49) | 40 (34-45) | 0.21 |
| Triglycerides, mg/dL | 130 (92-209) | 133 (89-174) | 0.64 |
| Non HDL-c, mg/dL | 155 (135-193) | 158 (132-191) | 0.84 |
| Creatinine, mg/dL | 0.88 (0.75-1.01) | 0.85 (0.75-1.02) | 0.93 |
| GFR, ml/min/1.73 m$^2$ | 88 (73-100) | 92 (79-100) | 0.72 |

Continuous variables are median (interquartiles) examined by Mann-Whitney test and qualitative variables are n (%) evaluated by the Pearson's Chi-square test. BMI – body mass index; hsTNT – high-sensitivity Troponin T; SBP – systolic blood pressure; DBP – diastolic blood pressure; GFR – glomerular filtration rate.

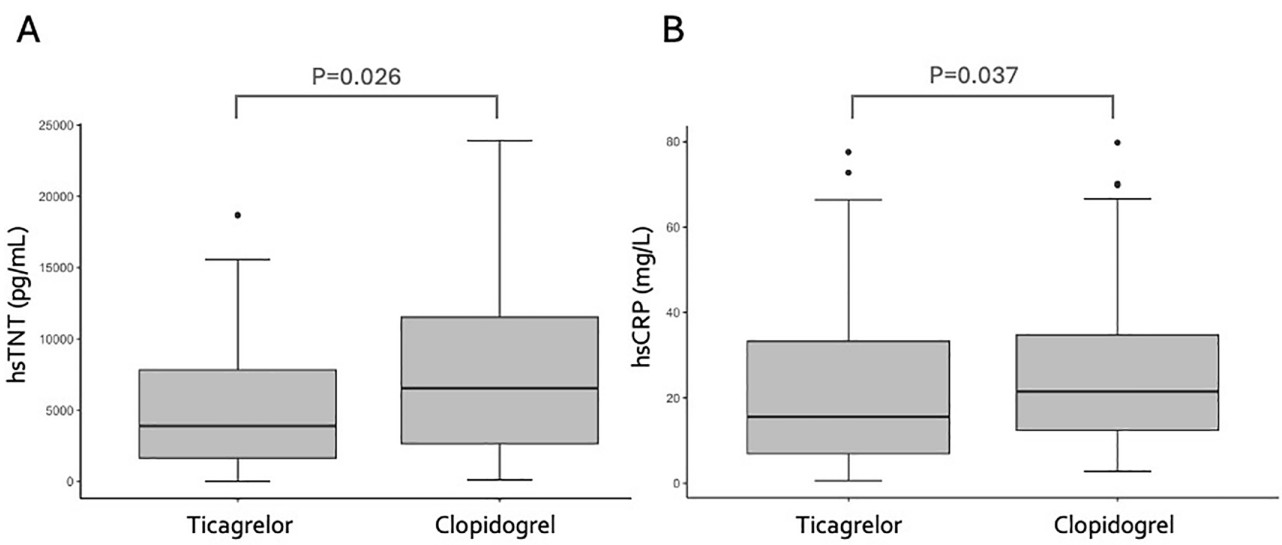

**Fig 2. Box plots illustrating differences between the ticagrelor and clopidogrel groups on the first day of myocardial infarction.** A – High-sensitivity troponin T (hsTNT); B – High-sensitivity C-reactive protein (hsCRP). Lower levels of hsTNT and hsCRP were observed among patients treated with ticagrelor compared to those treated with clopidogrel (Mann-Whitney test).

**Table 2. Inflammatory markers by treatment groups at baseline and 30 days.**

| | Ticagrelor | Clopidogrel | P value |
|---|---|---|---|
| **At baseline** | | | |
| N/L ratio | 4.5 (3.1-7.1) | 4.7 (3.2-7.0) | 0.99 |
| HsCRP, mg/L | 17.8 (7.0-40.3) | 23.4 (12.7-47.2) | 0.04 |
| TCD4, cells/mL | 960 (611-1294) | 869 (558-1228) | 0.51 |
| TCD8, cells/mL | 352 (243-442) | 315 (212-533) | 0.83 |
| B1, cells/mL | 4.7 (2.1-10.2) | 4.1 (2.5-7.8) | 0.67 |
| B2 mem, cells/mL | 57 (28-122) | 48 (24-110) | 0.74 |
| B2 naïve, cells/mL | 65 (14-118) | 48 (17-102) | 0.99 |
| B2 classic, cells/mL | 138 (87-260) | 124 (71-221) | 0.55 |
| IL-1b, pg/mL | 0.06 (0.00-0.12) | 0.05 (0.03-0.12) | 0.54 |
| IL-4, pg/mL | 3.7 (0.0-8.5) | 1.7 (0.0-8.1) | 0.27 |
| IL-6, pg/mL | 6.2 (0.0-20.0) | 7.1 (0.0-29.0) | 0.48 |
| IL-10, pg/mL | 4.5 (0.0-11.8) | 3.3 (0.2-10.1) | 0.76 |
| IL-18, pg/mL | 1145 (696-1600) | 1123 (872-1653) | 0.55 |
| **At 30 days** | | | |
| N/L ratio | 2.8 (2.4-3.6) | 2.9 (2.2-3.7) | 0.73 |
| HsCRP, mg/L | 2.1 (1.1-4.3) | 2.3 (0.9-6.4) | 0.92 |
| TCD4, cells/mL | 841 (685-1095) | 886 (656-1203) | 0.83 |
| TCD8, cells/mL | 329 (212-506) | 282 (198-498) | 0.82 |
| B1, cells/mL | 4.1 (2.5-9.1) | 3.0 (1.9-7.1) | 0.09 |
| B2 mem, cells/mL | 48 (26-96) | 38 (20-81) | 0.28 |
| B2 naïve, cells/mL | 41 (14-79) | 41 (11-87) | 0.94 |
| B2 classic, cells/mL | 108 (67-172) | 100 (52-149) | 0.49 |
| IL-1b, pg/mL | 0.02 (0.00-0.09) | 0.02 (0.00-0.11) | 0.75 |
| IL-4, pg/mL | 6.7 (3.5-10.7) | 6.0 (2.1-9.8) | 0.53 |
| IL-6, pg/mL | 11.2 (8.9-15.6) | 12.1 (8.8-14.9) | 0.96 |
| IL-10, pg/mL | 17.2 (7.3-26.5) | 16.9 (5.3-30.9) | 0.88 |
| IL-18, pg/mL | 351 (207-482) | 346 (216-463) | 0.80 |

Values are median (interquartiles). N/L – neutrophil/lymphocyte; hsCRP – high-sensitivity C-reactive protein; B2 mem – B2 memory; B2 classic (B2 naïve plus B2 memory cells). Variables were tested by the Mann-Whitney U test.

circumflex (LCx). Main findings of the coronary angiography and PCI are shown in Table 3. There were no differences between arms according to the Syntax or Gensini scores. In addition, there were similar TIMI flow grades pre- or post-PCI, predominantly TIMI flow grade 3 (Table 3).

Most patients achieved myocardial blush grade 3 in both arms, and few had collateral vessels, with no differences between arms by the Rentrop score (Table 3).

## Cardiac magnetic resonance outcomes

A smaller amount of infarcted mass was observed in the ticagrelor arm compared with the clopidogrel arm in grams or the percentage of left ventricular mass (Table 4). Better right ventricular ejection fraction (p = 0.044) and a trend towards better LVEF (p = 0.051) were obtained with the ticagrelor group (Table 4). Ticagrelor and clopidogrel groups had similar left ventricular mass. Sensitivity analysis using imputed values for patients who died did not alter the results.

**Table 3. Major angiographic characteristics before and after PCI.**

| | Ticagrelor | clopidogrel | P value |
|---|---|---|---|
| SYNTAX score | 7 (5-11) | 6 (4-11) | 0.49 |
| Gensini score | 10 (4-20) | 9 (4-20) | 0.74 |
| TIMI flow before (n) | | | 0.63 |
| 0 | 11 | 16 | |
| 1 | 4 | 3 | |
| 2 | 17 | 19 | |
| 3 | 61 | 53 | |
| TIMI flow after (n) | | | 0.61 |
| 0 | 1 | 1 | |
| 1 | 1 | 1 | |
| 2 | 9 | 15 | |
| 3 | 74 | 84 | |
| Blush (n) | | | 1.00 |
| 0 | 1 | 1 | |
| 1 | 1 | 1 | |
| 2 | 11 | 11 | |
| 3 | 74 | 72 | |
| Rentrop (n) | | | 0.51 |
| 0 | 89 | 87 | |
| 1 | 2 | 1 | |
| 2 | 0 | 1 | |
| Target artery or no PCI (n) | | | 0.70 |
| LAD | 45 | 44 | |
| RCA | 36 | 39 | |
| LCx | 10 | 8 | |
| No obstructive lesion | 12 | 6 | |
| Three-vessel disease | 4 | 4 | |
| Thrombus (n) | | | 0.82 |
| Absent | 47 | 44 | |
| Present | 46 | 46 | |
| Calcium (n) | | | 0.94 |
| Absent | 83 | 80 | |
| Present | 10 | 10 | |
| Coronary dissection (n) | | | 0.35 |
| Absent | 93 | 87 | |
| Present | 0 | 2 | |

n-number; PCI – percutaneous coronary intervention; LAD – left anterior descending; RCA – right coronary artery; LCx – Left circumflex artery. Syntax and Gensini scores were compared by the Mann-Whitney U test, and categorical variables by the Pearson Chi-square test or Fisher's exact test.

The exploratory correlations with myocardial fibrosis on the 1st day and 30th day are shown by the network graph in the ticagrelor and clopidogrel arms (Fig 3). The network graph shows the correlations with LVEF by treatments (Fig 4).

The K-means algorithm shows data obtained for fibrosis (%) and LVEF in clusters for ticagrelor and clopidogrel (Fig 5). The accuracy obtained was 57%, showing, in the ticagrelor arm, more patients displaying lower infarcted mass and higher LVEF.

## Clinical Outcomes

The study was not powered to evaluate clinical events. During the first 30 days after STEMI, there were four patients hospitalized with the need for urgent PCI, two patients with recurrent myocardial infarction, and one patient hospitalized due to heart failure. Twelve patients were submitted to elective CABG and 24 to elective PCI. During the first month post-STEMI, eight patients died (three in the ticagrelor arm and five in the clopidogrel arm). No differences between arms were observed for these clinical events (Table 5). The trial was stopped shortly before reaching the planned sample size in March 2020 due to the COVID-19 pandemic.

## Discussion

The main contribution of this study was to show that antiplatelet therapy with ticagrelor compared with clopidogrel, in a randomized clinical trial, was associated with smaller infarcted mass measured by CMR 30 days after STEMI. The study also revealed that patients treated with ticagrelor had more homogeneous responses towards smaller infarcted mass and better LVEF.

All patients were randomized to receive antiplatelet therapy before invasive procedures, within an appropriate window for thrombolytic therapy (less than six hours from symptom onset). Additionally, coronary angiography followed by PCI was performed within the first 24 hours of STEMI. A third key aspect of the study was the adoption of rigorous weekly post-discharge medical follow-up aimed at optimizing pharmacological treatment. The study was designed with the hypothesis that ticagrelor, by reducing adenosine uptake into cells, could enhance microcirculation and promote the recovery of ischemic cardiomyocytes [24]. Indeed, improvement of coronary microcirculation with ticagrelor has been previously demonstrated in patients undergoing elective PCI, using continuous thermodilution to estimate coronary blood flow and microvascular resistance at baseline and after antiplatelet therapy [36]. The lower hsTNT levels observed on the first day of STEMI in the ticagrelor group suggest an early benefit on microcirculation, as both study arms had a similar distribution of culprit arteries, Syntax, and Gensini scores after randomization. Furthermore, post-PCI assessments, including TIMI flow, myocardial blush, and Rentrop scores, were comparable between the two groups. Collectively, early improvement in microcirculation may have contributed, at least in part, to the reduction in myocardial injury.

A decrease in the intensity of inflammation has been proposed as a possible therapeutic target for reducing cardiovascular risk [14,15]. Inflammatory markers have been associated with infarct size and ventricular function following STEMI [26,27]. In our study, patients randomized to the clopidogrel arm exhibited higher hsCRP levels measured on the first day of STEMI. A previous study investigated inflammatory markers (hsCRP, interleukin-6, and endothelial cell-specific molecule 1) in STEMI patients randomized to treatment with clopidogrel or ticagrelor. These markers were assessed at admission, 24 hours, day 4, and day 7 post-STEMI. The authors reported that, compared with clopidogrel, ticagrelor reduced

**Table 4. Main cMRI parameters at 30 days after STEMI.**

|  | Ticagrelor group (n = 114) | Clopidogrel group (n = 111) | P value |
|---|---|---|---|
| Myocardial fibrosis, g | 12 (6-23) | 17 (9-28) | 0.012 |
| Myocardial fibrosis, % of LV | 11 (6-22) | 16 (9-27) | 0.008 |
| Left ventricular mass, g | 104 (83-121) | 103 (85-124) | 0.594 |
| LVEF, % | 51 (43-59) | 47 (38-58) | 0.051 |
| RVEF, % | 57 (51-66) | 55 (50-61) | 0.044 |

Variables are median (interquartiles); g – grams; LV – left ventricle; LVEF – left ventricular ejection fraction; RVEF – right ventricular ejection fraction. Variables were compared by the Mann-Whitney U test.

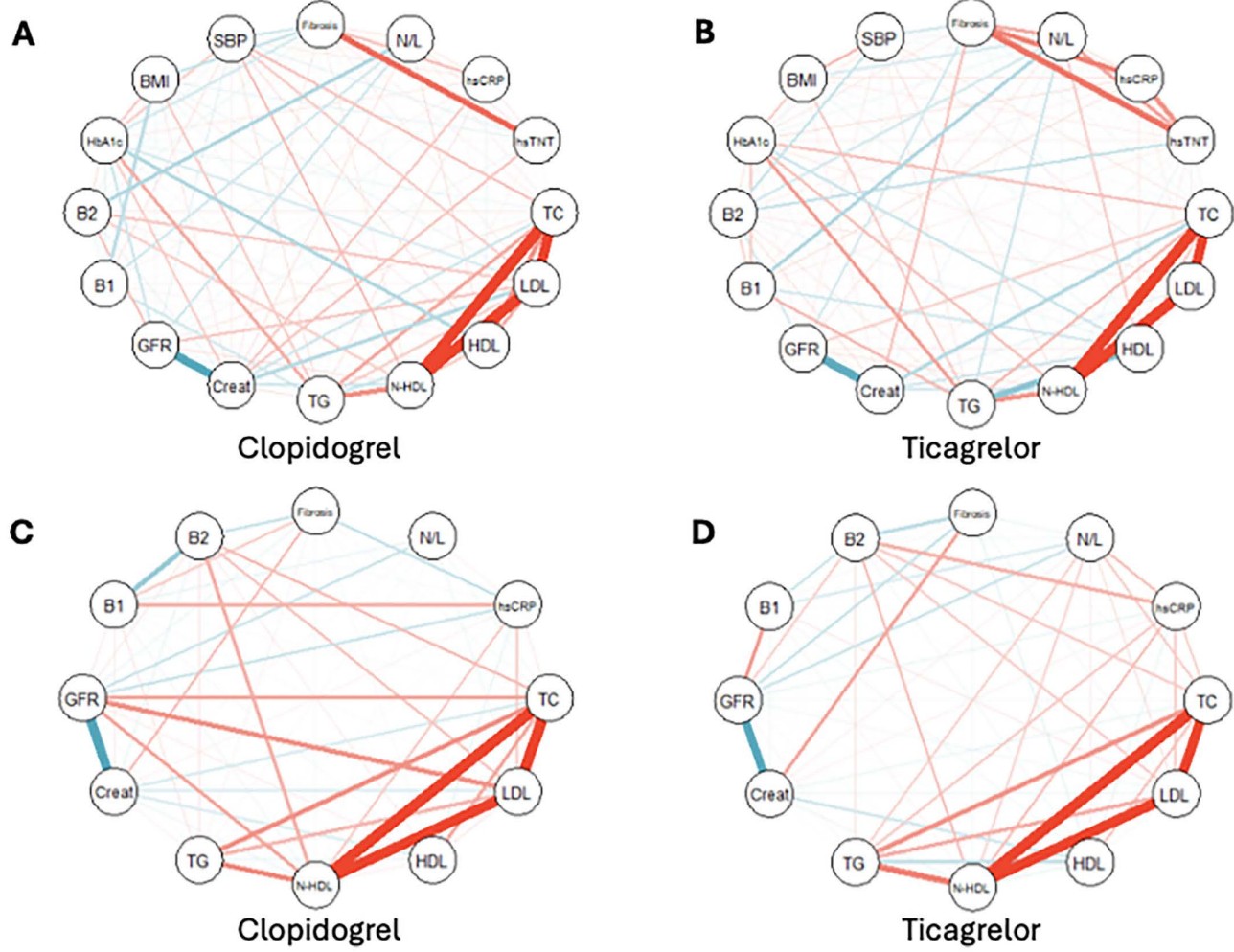

**Fig 3. Network graph showing variable correlations on the 1st day (A, B) and the 30th day (C, D).** In the ticagrelor arm, the most significant correlations with myocardial fibrosis on day 1 were high-sensitivity troponin T (hsTNT) and high-sensitivity C-reactive protein (hsCRP), whereas in the clopidogrel arm, hsTNT was the primary variable. By day 30, hsCRP, B2 cells, and creatinine were the main variables correlated with fibrosis in the clopidogrel arm, while creatinine and B2 cells were predominant in the ticagrelor arm. Blue lines indicate negative correlations; red lines indicate positive correlations.

these inflammatory markers more rapidly [37]. Thus, the differences observed in hsCRP levels between the ticagrelor and clopidogrel groups in our study may be attributed to a faster anti-inflammatory response with ticagrelor following STEMI [37]. A lower inflammatory state could also contribute to a smaller infarct size observed in the ticagrelor arm.

More effective lipid-lowering therapy has been consistently associated with clinical benefits and stabilization of atherosclerotic plaques [38]. In our study, all patients received high-intensity lipid-lowering therapy, with comparable standard lipid panels between arms at baseline and after 30 days. However, a previous study of our group explored the effects of antiplatelet therapy on LDL particle quality using Gaussian laser beam (Z-scan), small-angle X-ray scattering (SAXS), dynamic light scattering (DLS), ultraviolet-visible spectroscopy, and polyacrylamide gel electrophoresis. These methods assessed oxidative status, structural changes, particle size, and LDL subfractions, revealing better LDL quality in samples from patients treated with ticagrelor compared to clopidogrel [39]. The affinity of LDL receptors is influenced by LDL

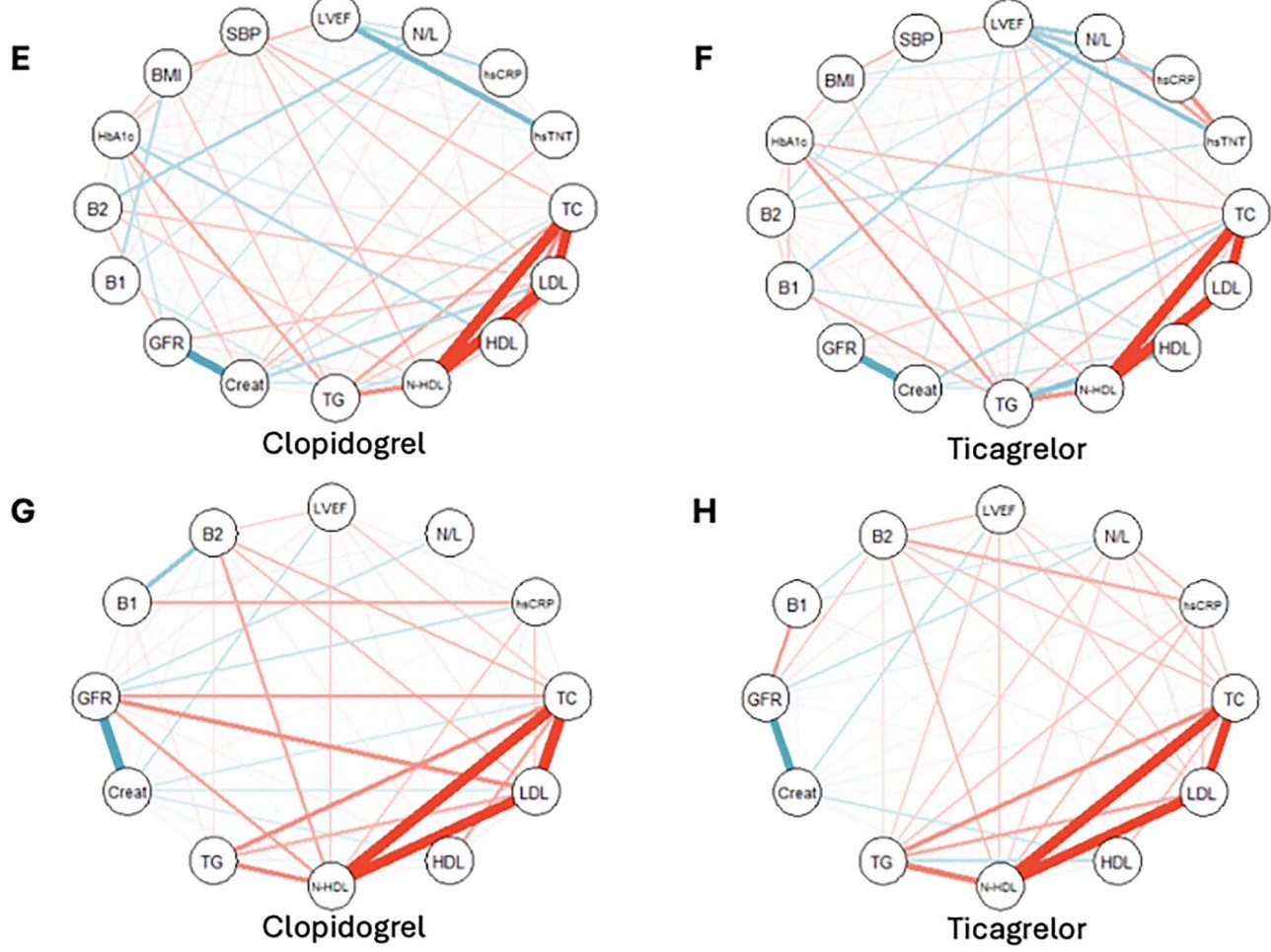

**Fig 4. Network graph depicting the most important correlations with left ventricular ejection fraction (LVEF).** On the 1st day **(E, F)**, in the clopidogrel arm, key correlations included high-sensitivity troponin T (hsTNT), high-sensitivity C-reactive protein (hsCRP), neutrophil-to-lymphocyte ratio (N/L), and systolic blood pressure (SBP), while in the ticagrelor arm, the same variables, along with lipid parameters, were correlated with LVEF. By the 30th day **(G, H)**, in the clopidogrel arm, creatinine, B2 cells, and lipid parameters were the main correlations with LVEF, whereas in the ticagrelor arm, B2 cells, creatinine, and lipid parameters were also predominant. Blue lines indicate negative correlations; red lines indicate positive correlations.

particle size and oxidation status, and the improved LDL quality following ticagrelor therapy may have long-term relevance, not seen at short-term with standard lipid profile. In the PLATO study, the primary endpoint (cardiovascular death, non-fatal myocardial infarction, or non-fatal stroke) was significantly reduced even in the period after the first 30 days up to one year [1]. These data suggest continued benefits in addition to those directly related to thrombosis or microcirculation during the acute phase of myocardial infarction. Our results were obtained in patients undergoing pharmaco-invasive strategy, the only group of acute coronary syndrome patients not included in the PLATO study [1]. Later, in the TREAT trial [10], the safety for bleeding in patients following pharmaco-invasive strategy was similar between those receiving ticagrelor or clopidogrel. However, the study did not have sufficient power to analyze major cardiovascular endpoints [10]. Therefore, for patients undergoing pharmaco-invasive strategy, despite promising data in our study, larger studies addressing cardiovascular outcomes are needed. More recently, DAPT strategies and de-escalation after 1–3 months have been examined. First, a comparison of six antiplatelet strategies in ACS patients undergoing PCI was reported, showing lower

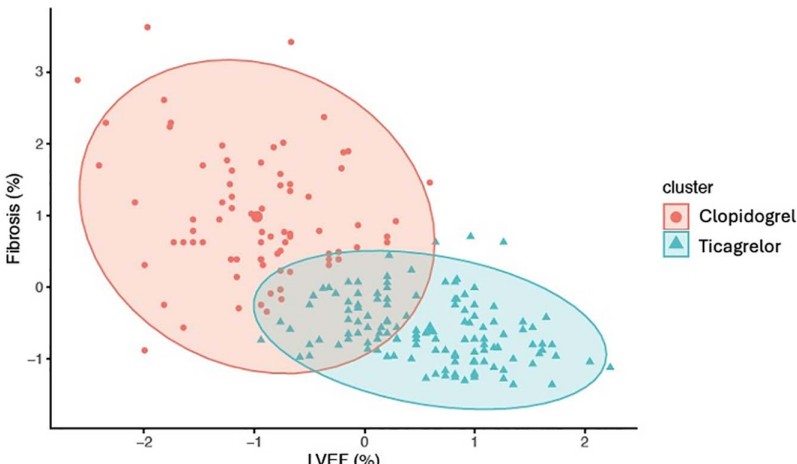

**Fig 5. Clusters of fibrosis and LVEF in the clopidogrel and ticagrelor arms.** K-means analysis demonstrates moderate accuracy (57.3%). The graph shows that most patients in the ticagrelor arm had better LVEF and smaller infarct size.

**Table 5. Cardiovascular outcomes at 30 days after STEMI.**

|  | Ticagrelor | Clopidogrel | P value |
|---|---|---|---|
| Death | 3 | 5 | 0.49 |
| Hospitalization due to unstable angina | 3 | 3 | 1.00 |
| Urgent Percutaneous coronary intervention | 3 | 1 | 0.62 |
| Recurrent myocardial infarction | 1 | 1 | 1.00 |
| Elective PCI | 12 | 12 | 0.94 |
| Elective CABG | 4 | 8 | 0.25 |
| Hospitalization due to heart failure | 1 | 0 | 1.00 |
| Bleeding | 6 | 5 | 1.00 |
| Sudden death resuscitation | 4 | 2 | 0.68 |

Data were compared by the Fisher's exact test. CABG – Coronary Artery Bypass Graft surgery.
PCI – percutaneous coronary intervention.

bleeding rates with de-escalation of DAPT without increase in stent thrombosis [40]. Further, meta-analysis of six trials showed that compared with 12-months of DAPT, de-escalation (after 1–3 months of DAPT), with ticagrelor as a single antiplatelet agent, reduced bleeding rates and did not increase ischemic events. In fact, DAPT de-escalation was associated with decrease in cardiovascular and all-cause mortality [41].

## Study limitations and strengths

The BATTLE-AMI trial included only STEMI patients under 75 without previous myocardial infarction on a pharmaco-invasive strategy. Therefore, caution is needed when extrapolating these findings to patients treated by primary PCI. However, these findings were obtained in patients with timely coronary reperfusion receiving optimal medical treatment (including beta-blockers, renin-angiotensin system blockers, and lipid-lowering therapies). The study sample size was relatively small, but the infarct size was measured by CMR, with both arms exhibiting a comparable degree of coronary disease, type of culprit artery, and the same success rate for reperfusion.

## Conclusion

In STEMI patients managed with a pharmaco-invasive strategy, receiving optimal medical therapy and timely invasive procedures, those treated with ticagrelor exhibited smaller infarct sizes compared to those receiving clopidogrel, despite having similar pre- and post-reperfusion angiographic coronary characteristics. Larger studies are warranted to validate these findings.

## Supporting information

**S1 File. Laboratory parameters at 30 days after STEMI.** Laboratory parameters did not differ between antiplatelet groups at 30 days after STEMI.
(DOCX)

**S2 File. The B And T Types of Lymphocytes Evaluation in Acute Myocardial Infarction (BATTLE-AMI) trial investigators.** This project aimed to compare the effects of lipid-lowering and antiplatelet therapies (2x2 factorial design) on the infarct size and ventricular remodeling (CMR). Researchers from *Universidade Federal de São Paulo*, *Universidade Estadual de Campinas*, *Universidade Santo Amaro*, *Universidade São Paulo*, *Instituto Dante Pazzanese de Cardiologia*, *Hospital Israelita Albert Einstein*, Royal Imperial College (UK), participated in the study.
(DOCX)

**S3 File. CONSORT 2010 checklist.** The main information of each section of the randomized clinical trial is reported on corresponding pages.
(PDF)

## Acknowledgments

We thank Mayara M Garcia (BioStats, Brazil) for the statistical analysis of the study, and the BATTLE-AMI group (names and affiliations in supplementary file).

## Author contributions

**Conceptualization:** Francisco A Fonseca, Gilberto Szarf, Ibraim Pinto, Carolina N França, Amanda S Bacchin, Maria C Izar.

**Data curation:** Francisco A Fonseca, Adriano Caixeta, Gilberto Szarf, Ibraim Pinto, Antonio M Figueiredo Neto, Henrique T Bianco, Michelle Birtche, Igor R M Batista, Maria C Izar.

**Formal analysis:** Francisco A Fonseca, Adriano Caixeta, Gilberto Szarf, Ibraim Pinto, Antonio M Figueiredo Neto, Carolina N França, Henrique T Bianco, Henrique A Fonseca, Michelle Birtche, Igor R M Batista, Maria C Izar.

**Funding acquisition:** Francisco A Fonseca, Antonio M Figueiredo Neto.

**Investigation:** Francisco A Fonseca, Adriano Caixeta, Gilberto Szarf, Ibraim Pinto, Antonio M Figueiredo Neto, Carolina N França, Henrique T Bianco, Amanda S Bacchin, Michelle Birtche, Igor R M Batista, Maria C Izar.

**Methodology:** Francisco A Fonseca, Adriano Caixeta, Gilberto Szarf, Ibraim Pinto, Antonio M Figueiredo Neto, Carolina N França, Henrique A Fonseca, Michelle Birtche, Igor R M Batista, Maria C Izar.

**Project administration:** Francisco A Fonseca, Maria C Izar.

**Resources:** Francisco A Fonseca.

**Software:** Francisco A Fonseca, Adriano Caixeta, Gilberto Szarf, Michelle Birtche, Igor R M Batista.

**Supervision:** Francisco A Fonseca, Adriano Caixeta, Gilberto Szarf, Ibraim Pinto, Antonio M Figueiredo Neto, Carolina N França, Henrique T Bianco, Henrique A Fonseca, Amanda S Bacchin, Michelle Birtche, Maria C Izar.

**Validation:** Francisco A Fonseca, Adriano Caixeta, Gilberto Szarf, Ibraim Pinto.

**Visualization:** Francisco A Fonseca.

**Writing – original draft:** Francisco A Fonseca.

**Writing – review & editing:** Francisco A Fonseca, Adriano Caixeta, Gilberto Szarf, Ibraim Pinto, Antonio M Figueiredo Neto, Carolina N França, Henrique T Bianco, Henrique A Fonseca, Amanda S Bacchin, Michelle Birtche, Igor R M Batista, Maria C Izar.

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
