## [Decision Letter · Decision Letter 0]

30 Jul 2025

Dear Dr. Fonseca,

Thank you for submitting your manuscript to PLOS ONE. After careful consideration, we feel that it has merit but does not fully meet PLOS ONE’s publication criteria as it currently stands. Therefore, we invite you to submit a revised version of the manuscript that addresses the points raised during the review process.

We look forward to receiving your revised manuscript.

Kind regards,

Shukri AlSaif

Academic Editor

PLOS ONE

Journal Requirements: 

2. We note that you have selected “Clinical Trial” as your article type. PLOS ONE requires that all clinical trials are registered in an appropriate registry (the WHO list of approved registries is at https://www.who.int/clinical-trials-registry-platform/network/primary-registries " https://www.who.int/clinical-trials-registry-platform/network/primary-registries and more information on trial registration is at http://www.icmje.org/about-icmje/faqs/clinical-trials-registration/ ). Please state the name of the registry and the registration number (e.g. ISRCTN or ClinicalTrials.gov ) in the submission data and on the title page of your manuscript. a) Please provide the complete date range for participant recruitment and follow-up in the methods section of your manuscript. b) If you have not yet registered your trial in an appropriate registry, we now require you to do so and will need confirmation of the trial registry number before we can pass your paper to the next stage of review. Please include in the Methods section of your paper your reasons for not registering this study before enrolment of participants started. Please confirm that all related trials are registered by stating: “The authors confirm that all ongoing and related trials for this drug/intervention are registered”. Please see http://journals.plos.org/plosone/s/submission-guidelines#loc-clinical-trials for our policies on clinical trials.

4. One of the noted authors is a group or consortium [BATTLE-AMI investigators]. In addition to naming the author group, please list the individual authors and affiliations within this group in the acknowledgments section of your manuscript. Please also indicate clearly a lead author for this group along with a contact email address.

Reviewers' comments:

Reviewer's Responses to Questions

**Comments to the Author**

1. Is the manuscript technically sound, and do the data support the conclusions?

Reviewer #1: Yes

Reviewer #2: Yes

2. Has the statistical analysis been performed appropriately and rigorously?

Reviewer #1: Yes

Reviewer #2: Yes

3. Have the authors made all data underlying the findings in their manuscript fully available?

Reviewer #1: Yes

Reviewer #2: Yes

4. Is the manuscript presented in an intelligible fashion and written in standard English?

Reviewer #1: Yes

Reviewer #2: Yes

Reviewer #1: The manuscript presents a well-designed randomized clinical trial comparing the effects of ticagrelor and clopidogrel on infarct size in STEMI patients managed with a pharmaco-invasive strategy. The manuscript is technically robust. The study design adheres to rigorous clinical trial standards. The data strongly support the primary conclusion that ticagrelor is associated with a smaller infarct size compared to clopidogrel in STEMI patients, including the primary outcome of a significantly smaller percentage of left ventricular infarcted mass (p=0.012), with consistent results in both absolute (grams) and relative (%) measurements. Despite similar baseline angiographic characteristics (Syntax score, TIMI flow, culprit artery distribution) and post-PCI reperfusion success, the ticagrelor group showed more homogeneous responses (via K-means clustering) with smaller infarct sizes and better LVEF. Lower high-sensitivity troponin T (hsTNT) and high-sensitivity C-reactive protein (hsCRP) on day 1 in the ticagrelor group suggest early benefits on microcirculation, reinforcing the mechanistic link to reduced infarct size. Confounding variables (e.g., lipid profiles, other inflammatory markers) were balanced between groups, strengthening the causal inference. The statistical approach is rigorous and appropriate for the study design. The manuscript is well-structured and written in clear, standard English. Tables and figures are informative and support the narrative, though legends could be slightly expanded to clarify statistical significance markers for non-specialist readers.

The study is limited to STEMI patients <75 years without prior myocardial infarction, managed via a pharmaco-invasive strategy. Caution is advised when extrapolating to primary PCI populations, which the authors appropriately note in limitations. The discussion links ticagrelor's benefits to improved microcirculation (via adenosine uptake inhibition) and reduced inflammation, supported by prior literature. Expanding on how improved LDL quality (cited in ref. 39) might contribute to long-term outcomes could strengthen the mechanistic narrative. While the sample (n=225) is sufficient to detect differences in infarct size, larger trials are warranted to validate these findings, as acknowledged in the conclusion.

This manuscript is methodologically sound, with robust data supporting its conclusions. The rigorous design, appropriate use of CMR, and thorough statistical analysis make it a valuable contribution to understanding antiplatelet therapy in STEMI. Minor refinements to mechanistic discussions and generalizability notes would further enhance its impact. We recommend consideration for publication.

Reviewer #2: Interesting paper. Some issues should be addresed

Major

Do authors think that these fidngins hold true for patients treated with pPCI without thrombolysis?

Time of enrollement compared to number of patients appears long. Please comment

Abstract: please add numeric values for the area at risk

Recently lenght and kind of DAPT demonstrated to change prognosis. Please relate your findings with those from a clinical point of view (quote on PMID: 38242567)

**Do you want your identity to be public for this peer review?** For information about this choice, including consent withdrawal, please see our Privacy Policy

Reviewer #1: **Yes: ** Dong Huang

Reviewer #2: **Yes: ** Fabrizio D'Ascenzo

---

## [Author Response · Author response to Decision Letter 1]

27 Aug 2025

Responses to reviewers

Reviewer #1:

The manuscript presents a well-designed randomized clinical trial comparing the effects of ticagrelor and clopidogrel on infarct size in STEMI patients managed with a pharmaco-invasive strategy. The manuscript is technically robust. The study design adheres to rigorous standards for clinical trials. The data strongly support the primary conclusion that ticagrelor is associated with a smaller infarct size compared to clopidogrel in STEMI patients, including the primary outcome of a significantly smaller percentage of left ventricular infarcted mass (p = 0.012), with consistent results in both absolute (grams) and relative (%) measurements. Despite similar baseline angiographic characteristics (Syntax score, TIMI flow, culprit artery distribution) and post-PCI reperfusion success, the ticagrelor group showed more homogeneous responses (via K-means clustering) with smaller infarct sizes and better LVEF. Lower high-sensitivity troponin T (hsTNT) and high-sensitivity C-reactive protein (hsCRP) on day 1 in the ticagrelor group suggest early benefits on microcirculation, reinforcing the mechanistic link to reduced infarct size. Confounding variables (e.g., lipid profiles, other inflammatory markers) were balanced between groups, strengthening the causal inference. The statistical approach is rigorous and appropriate for the study design. The manuscript is well-structured and written in clear, standard English. Tables and figures are informative and support the narrative, though legends could be slightly expanded to clarify statistical significance markers for non-specialist readers.

Authors: We added information in the legend of figure 2.

Figure 2. Box plots illustrating differences between the ticagrelor and clopidogrel groups on the first day of myocardial infarction. A – High-sensitivity troponin (hsTNT); B – High-sensitivity C-reactive protein (hsCRP). Lower levels of hsTNT and hsCRP were observed among patients treated with ticagrelor compared to those treated with clopidogrel (Mann-Whitney test).

The study is limited to STEMI patients <75 years without prior myocardial infarction, managed via a pharmaco-invasive strategy. Caution is advised when extrapolating to primary PCI populations, which the authors appropriately note in limitations. The discussion links ticagrelor's benefits to improved microcirculation (via adenosine uptake inhibition) and reduced inflammation, supported by prior literature. Expanding on how improved LDL quality (cited in ref. 39) might contribute to long-term outcomes could strengthen the mechanistic narrative. While the sample (n=225) is sufficient to detect differences in infarct size, larger trials are warranted to validate these findings, as acknowledged in the conclusion.

This manuscript is methodologically sound, with robust data supporting its conclusions. The rigorous design, appropriate use of CMR, and thorough statistical analysis make it a valuable contribution to understanding antiplatelet therapy in STEMI. Minor refinements to mechanistic discussions and generalizability notes would further enhance its impact. We recommend consideration for publication.

Authors: The authors appreciate your comments. We added minor refinements to mechanistic discussions and generalizability notes to improve the manuscript (see page 10, lines 352-359).

In the PLATO study, the primary endpoint (cardiovascular death, non-fatal myocardial infarction, or non-fatal stroke) was significantly reduced even in the period after the first 30 days up to one year1. These data suggest continued benefits in addition to those directly related to thrombosis or microcirculation during the acute phase of myocardial infarction. Our results were obtained in patients undergoing a pharmaco-invasive strategy, the only group of acute coronary syndrome patients not included in the PLATO study1.

Reviewer #2:

Interesting paper. Some issues should be addressed

Major

Do authors think that these findings hold true for patients treated with pPCI without thrombolysis?

Time of enrollment compared to number of patients appears long. Please comment

Abstract: please add numeric values for the area at risk

Recently length and kind of DAPT demonstrated to change prognosis. Please relate your findings with those from a clinical point of view (quote on PMID: 38242567)

Authors: We appreciate your comments.

Do authors think that these findings hold true for patients treated with pPCI without thrombolysis?

Authors: It is an important question (see page 10 lines 352-362).

In the PLATO study, the primary endpoint (cardiovascular death, non-fatal myocardial infarction, or non-fatal stroke) was significantly reduced even in the period after the first 30 days up to one year1. These data suggest continued benefits in addition to those directly related to thrombosis or microcirculation during the acute phase of myocardial infarction. Our results were obtained in patients undergoing pharmaco-invasive strategy, the only group of acute coronary syndrome patients not included in the PLATO study1. Later, in the TREAT trial10, the safety for bleeding in patients following pharmaco-invasive strategy was similar between those receiving ticagrelor or clopidogrel. However, the study did not have sufficient power to analyze major cardiovascular endpoints10. Therefore, for patients undergoing pharmaco-invasive strategy, despite promising data in our study, larger studies addressing cardiovascular outcomes are needed.

Time of enrollment compared to number of patients appears long. Please comment

Authors: In fact, we included patients from May 2015 to March 2020. This study is part of the BATTLE-AMI trial, a thematic study addressing the key players in the amount of infarcted mass and LVEF quantified by CMR. In the trial protocol we included only stable patients undergoing pharmacological thrombolysis in the first 6 hours of symptoms onset and referred to our university hospital in the first 24 hours for coronary angiography and invasive procedures. Based on the inclusion and exclusion criteria, we gradually reached approximately the planned number of patients only in March 2020. At this point, we had the challenge of the coronavirus epidemic and we decided to end the study. (see pages 3 lines 99-126).

Abstract: please add numeric values for the area at risk

Recently length and kind of DAPT demonstrated to change prognosis. Please relate your findings with those from a clinical point of view (quote on PMID: 38242567)

Authors: unfortunately we did not measure the area at risk by CMR, however the groups were comparable for both pre- and post-PCI, and for most clinical and laboratory data. Area at risk is an important topic and we will include it in future manuscripts using a new software for this analysis.

The length and kind of DAPT is really a very important issue. We added two recent meta-analyses addressing bleeding rates and ischemic events among ACS patients with de-escalation of DAPT. (see page 10,11, lines 364-372; ref 40 and 41).

More recently, DAPT strategies and de-escalation after 1-3 months have been examined. First, a comparison of six antiplatelet strategies in ACS patients undergoing PCI was reported, showing lower bleeding rates with de-escalation of DAPT without increase in stent thrombosis40. Further, meta-analysis of six trials showed that compared with 12-months of DAPT, de-escalation (after 1-3 months of DAPT), with ticagrelor as a single antiplatelet agent, reduced bleeding rates and did not increase ischemic events. In fact, DAPT de-escalation was associated with decrease in cardiovascular and all-cause mortality41.

The authors appreciate your valuable comments.

---

## [Decision Letter · Decision Letter 1]

14 Sep 2025

Smaller Infarct Size with Ticagrelor vs. Clopidogrel in STEMI Patients: Insights from Cardiac Magnetic Resonance

PONE-D-25-30655R1

Dear Dr.Francisco A H Fonseca,

We’re pleased to inform you that your manuscript has been judged scientifically suitable for publication and will be formally accepted for publication once it meets all outstanding technical requirements.

Kind regards,

Shukri AlSaif

Academic Editor

PLOS ONE

Additional Editor Comments (optional):

Reviewer #2:

Reviewers' comments:

Reviewer's Responses to Questions

**Comments to the Author**

Reviewer #2: All comments have been addressed

2. Is the manuscript technically sound, and do the data support the conclusions?

Reviewer #2: Yes

3. Has the statistical analysis been performed appropriately and rigorously?

Reviewer #2: Yes

4. Have the authors made all data underlying the findings in their manuscript fully available?

Reviewer #2: Yes

5. Is the manuscript presented in an intelligible fashion and written in standard English?

Reviewer #2: Yes

Reviewer #2: All comments have been addressed and authors should be complimented for addressing such a relevant issues.

**Do you want your identity to be public for this peer review?** For information about this choice, including consent withdrawal, please see our Privacy Policy

Reviewer #2: **Yes: ** Fabrizio D'Ascenzo

---

## [Editor Report · Acceptance letter]

PONE-D-25-30655R1

PLOS ONE

Dear Dr. Fonseca,

I'm pleased to inform you that your manuscript has been deemed suitable for publication in PLOS ONE. Congratulations! Your manuscript is now being handed over to our production team.

Kind regards,

on behalf of

Dr. Shukri AlSaif

Academic Editor

PLOS ONE